# Privacy-Aware Randomized Quantization via Linear Programming

**Zhongteng Cai**[1]  **Xueru Zhang**[1]  **Mohammad Mahdi Khalili**[1,2]

[1]Department of Computer Science and Engineering, Ohio State University, Columbus, Ohio, USA
[2] Yahoo Research, New York, New York, USA

## Abstract

Differential privacy mechanisms such as the Gaussian or Laplace mechanism have been widely used in data analytics for preserving individual privacy. However, they are mostly designed for continuous outputs and are unsuitable for scenarios where discrete values are necessary. Although various quantization mechanisms were proposed recently to generate discrete outputs under differential privacy, the outcomes are either biased or have an inferior accuracy-privacy trade-off. In this paper, we propose a family of quantization mechanisms that is unbiased and differentially private. It has a high degree of freedom and we show that some existing mechanisms can be considered as special cases of ours. To find the optimal mechanism, we formulate a linear optimization that can be solved efficiently using linear programming tools. Experiments show that our proposed mechanism can attain a better privacy-accuracy trade-off compared to baselines.

## 1 INTRODUCTION

Differential privacy (DP) has become a de facto standard for preserving individual data privacy in data analysis, ranging from simple tasks such as data collection and statistical analysis to complex machine learning tasks [Wang and Zhou, 2020, Jayaraman and Evans, 2019, Zhang et al., 2018a,b, 2019, 2022, Liu et al., 2021, Khalili et al., 2021b,a, Hopkins et al., 2022]. It centers around the idea that the output of a certain mechanism or computational procedure should be statistically similar given singular changes to the input, thereby preventing meaningful inference from observing the output. Although many DP mechanisms such as Gaussian mechanism [Dwork et al., 2014], Laplace mechanism [Dwork et al., 2006], etc., have been proposed to date to preserve individual privacy for different computational tasks,

they are mostly designed for continuous outputs over the real numbers and are unsuitable for scenarios where discrete outputs are necessary.

Indeed, keeping outputs discrete is desirable and even necessary for many applications. For example, representing real numbers on a finite computer requires data discretization, but naively using finite-precision rounding may compromise privacy [Mironov, 2012]. Real-valued outputs can induce high communication overheads, and compressing the continuous inputs to discrete and bounded outputs may be necessary for settings with bandwidth bottlenecks, e.g., federated learning [Reisizadeh et al., 2020, Jin et al., 2024]. Moreover, continuous outputs are incompatible with cryptographic primitives such as secure aggregation [Bonawitz et al., 2017]. It is thus essential to develop DP mechanisms that generate discrete outputs while preserving privacy.

To tackle the challenges mentioned above, many discrete DP mechanisms have been proposed, e.g., [Canonne et al., 2020, Agarwal et al., 2021, Kairouz et al., 2021]. However, the outputs generated by these mechanisms may be *biased* under truncation. Because in many applications such as machine learning, survey data collection, etc., it is often crucial to maintain the *unbiasedness* of private outputs, these approaches may not be desirable. For instance, when differentially private gradients are used to update machine learning models, keeping them unbiased helps the model gets updated towards the optimal solution and converges faster [Bottou et al., 2018].

To the best of our knowledge, only a few works proposed mechanisms that can generate discrete unbiased outputs under DP. This includes 1) *Minimum Variance Mechanism* (MVU) [Chaudhuri et al., 2022], which samples outputs from discrete alphabets and achieves the optimal utility by optimizing both the sampling probabilities and output alphabets. However, as the size of output alphabet increases, solving this optimization problem can be particularly challenging and the unbiasedness constraint must be relaxed; 2) *Randomized Quantization Mechanism* (RQM) [Youn et al., 2023]

which randomly maps inputs to closest pair of sampled bins. However, RQM assumes uniformly distributed bins and has only three hyperparameters that can be tuned, hence has smaller search space for hyperparameters to achieve good privacy-accuracy trade-off compared with MVU; 3) *Poisson Binomial Mechanism* (PBM) [Chen et al., 2022] which generates unbiased estimators by mapping inputs to a discrete distribution with bounded support. However, PBM has inferior flexibility and utility-privacy trade-off than RQM because it has fewer hyperparameters; 4) other DP mechanisms such as *Distributed Discrete Gaussian Mechanism* [Kairouz et al., 2021] and *Skellam Mechanism* [Agarwal et al., 2021] are unbiased on the unbounded support. However, they have to be truncated when combined with secure aggregation protocols, which will produce biased outputs.

This paper proposes a novel randomized quantization mechanism with discrete, unbiased outputs under DP guarantee. Importantly, our mechanism ensures unbiasedness regardless of the number of output bits; it is a general framework and the existing mechanism RQM can indeed be considered as a special case of ours. Specifically, given a set of quantization bins $B_1 < B_2 < \cdots < B_m$, discrete DP mechanism maps the continuous input $x$ to one of these bins. Our mechanism first samples two bins from the left and the right side of the input based on a pre-defined *selection distribution*, and then outputs one of the bins with unbiased expectation. For an example where $m = 4$ and $x \in [B_2, B_3)$. Our mechanism first randomly selects one bin on the left of $x$ (e.g., $B_1$) and another bin on the right (e.g., $B_3$) according to a selection distribution, then randomly outputs either $B_1$ or $B_3$ while preserving unbiasedness. The key is to carefully design selection distributions that maximize the accuracy of quantized outputs subject to DP constraint. Although this problem can be easily formulated as a non-linear constraint optimization, we propose a method that turns such non-linear optimization into a linear program that can be solved efficiently using linear programming tools. Experiments on both synthetic and real data validate the effectiveness of the proposed method.

Our contribution can be summarized as follows:

1. We propose a family of differentially private quantization mechanisms that generate discrete and unbiased outputs.

2. We theoretically quantify the privacy and accuracy of the exponential randomized mechanism (ERM), a special case of our proposed mechanism where selection distribution is based on DP exponential mechanism.

3. We design a linear program to find the optimal selection distribution of our mechanism, resulting in the optimal randomized quantization mechanism (OPTM), which attains a better accuracy-privacy trade-off.

4. We conduct experiments on various tasks to show our mechanisms, including both ERM and OPTM, attain superior performance than baselines.

## 2   RELATED WORKS

**Discrete differential privacy.**   Various discrete DP mechanisms have been proposed for discrete inputs to make them differentially private. For example, both *Discrete Laplace Mechanism* [Ghosh et al., 2009] and *Discrete Gaussian Mechanism* [Canonne et al., 2020] add noises to the inputs sampled from discrete distributions, which are commonly used for tasks when with discrete inputs [Abowd, 2018]. The *Snapping Mechanism* [Mironov, 2012] truncates and rounds the inputs and Laplace noises based on floating-point arithmetic, but it inevitably diminishes accuracy [Canonne et al., 2020]. *Communication-limited Local Differential Privacy* (CLDP) mechanism [Girgis et al., 2021] works with a trusted shuffler in federated learning to generate compressed and private updates from clients. However, it cannot be tuned to adopt different communication budgets. *Skellam Mechanism* [Agarwal et al., 2021] add noises sampled from Skellam distribution to achieve performance comparable with the continuous Gaussian mechanism, but is subject to biased output when combined with privacy-protection protocols in federated learning such as secure aggregation. In contrast, *Poisson Binomial Mechanism* [Chen et al., 2022] encodes the inputs inside the Binomial distribution to generate unbiased outputs, and it can achieve better privacy while decreasing communication costs, and is also compatible with secure aggregation.

**Private quantization.**   Previous works have utilized data compression methods such as quantization to compress the data in applications with communication or bandwidth bottlenecks. One example is federated learning where a central server needs to repeatedly collect local model updates from distributed clients for training the global model [Reisizadeh et al., 2020, Hönig et al., 2022]. Another example is large language models where the computation overheads may be reduced by compressing the model parameters [Tao et al., 2022, Bai et al., 2022]. By mapping the continuous inputs to the closest discrete outputs within a finite set, the quantization process can effectively represent the data with reduced communication overhead.

While methods were proposed in prior works to quantize data under a certain privacy constraint, they often treat privacy and quantization separately [Gandikota et al., 2021, Kairouz et al., 2021], i.e., privatizing the data first and then quantize the private data. Recent works attempt to design discrete DP mechanisms leveraging quantization to simultaneously compress data and protect privacy. For instance, Chaudhuri et al. [2022] proposed *Minimum Variance Mechanism* (MVU), which first quantizes inputs with discrete bins and then maps the unbiased quantization results to output alphabets according to a probability matrix. MVU optimizes the probability matrix to minimize accuracy loss while preserving privacy. l-MVU [Guo et al., 2023] extends MVU by designing a new interpolation procedure to attain better

privacy for high-dimensional vectors. Youn et al. [2023] proposed *Randomized Quantization Mechanism* (RQM), which subsamples from uniformly distributed bins and performs randomized quantization to output an unbiased result.

Compared to prior works, we propose a more general family of quantization DP mechanisms that enables non-uniform quantization. It has a high degree of freedom and the optimal mechanism can be found efficiently by linear programming tools. We also show theoretically and empirically that our mechanism can attain a better privacy-accuracy trade-off.

# 3  PROBLEM FORMULATION

Consider a quantization mechanism

$$\mathcal{M} : \mathcal{X} \to \{B_1, B_2, \cdots, B_m\}$$

used for quantizing a scalar $x \in [-c, c] := \mathcal{X}$, where

$$-c - \Delta = B_1 < B_2 < \cdots < B_m = c + \Delta,$$

$m$ is the number of quantization bins, $\Delta \geq 0$ extends the range of output. Note that $m$ bins here are not necessarily uniformly distributed. Our goal is to design $\mathcal{M}$ (including bin values $B_1, \cdots, B_m$ and $\Delta$) that is 1) differentially private; 2) unbiased, i.e., $\mathbb{E}(\mathcal{M}(x)) = x, \forall x$; and 3) accurate with the mean absolute error $\mathbb{E}(|\mathcal{M}(X) - X|)$ minimized. Let the capital letter $X$ denote the random variable of input and the small letter $x$ the corresponding realization.

## 3.1  BACKGROUND: DIFFERENTIAL PRIVACY

Differential privacy [Dwork, 2006], a widely used notion of privacy, ensures that no one by observing the computational outcome can infer a particular individual's data with high confidence. Formally, we say a randomized algorithm $\mathcal{M}(\cdot)$ satisfies $\epsilon$-differential privacy (DP) if for any two datasets $D$ and $D'$ that are different in at most one individual's data and for any set of possible outputs $S \subseteq \text{Range}(\mathcal{M})$, we have,

$$\Pr\{\mathcal{M}(D) \in S\} \leq \exp\{\epsilon\} \cdot \Pr\{\mathcal{M}(D') \in S\}.$$

where $\epsilon \in [0, \infty)$ is called privacy loss and can serve as a proxy for privacy leakage; the smaller $\epsilon$ implies a stronger privacy guarantee. Intuitively, for sufficiently small $\epsilon$, DP implies that the distribution of output remains almost the same if one individual's data changes in the dataset, and an attacker cannot reconstruct input data with high confidence after observing the output of mechanism $\mathcal{M}$.

Many mechanisms have been developed in the literature to satisfy differential privacy. One that is commonly used for scenarios with discrete outputs is *exponential mechanism* [McSherry and Talwar, 2007], as defined below.

**Definition 1 (Exponential Mechanism)** *Let the set of all possible outcomes of mechanism $\mathcal{M}$ be $\mathcal{O} = \{o_1, \cdots, o_{\hat{n}}\}$. Let $v : \mathcal{O} \times \mathcal{D} \to \mathbb{R}$ be a score function with a higher value of $v(o_i, D)$ indicating output $o_i$ is more desirable under dataset $D$. Let $\delta = \max_{i,D,D'} |v(o_i, D) - v(o_i, D')|$ be the sensitivity of score function, where $D$ and $D'$ are two datasets differing in at most one individual's data. Then, exponential mechanism $\mathcal{M} : \mathcal{D} \to \mathcal{O}$ that satisfies $\epsilon$-differential privacy selects $o_i \in \mathcal{O}$ with probability*

$$\Pr\{\mathcal{M}(D) = o_i\} = \frac{\exp\left\{\epsilon \cdot \frac{v(o_i, D)}{2\delta}\right\}}{\sum_{j=1}^{\hat{n}} \exp\left\{\epsilon \cdot \frac{v(o_j, D)}{2\delta}\right\}}.$$

## 3.2  PROPOSED QUANTIZATION MECHANISM

Next, we present our mechanism $\mathcal{M}$ that quantizes input with DP guarantee. Given a set of bins $\{B_1, B_2, \cdots, B_m\}$, $\mathcal{M}$ takes the following steps to quantize a scalar $x$:

1. For any $x$, select two bins $B_l, B_r \in \{B_1, B_2, \cdots, B_m\}$ randomly based on a pre-defined *selection distribution*, with $B_l \leq x$ located on the left side of $x$ and $B_r > x$ on the right side of $x$. In other words, if $x \in [B_j, B_{j+1})$, then $l \in \{1, \ldots, j\}$ and $r \in \{j + 1, \ldots, m\}$.

2. Then, $\mathcal{M}$ randomly outputs either $B_l$ or $B_r$ according to

$$\mathcal{M}(x) = \begin{cases} B_l, & \text{with probability (w.p.) } \frac{B_r - x}{B_r - B_l}; \\ B_r, & \text{with probability (w.p.) } \frac{x - B_l}{B_r - B_l}. \end{cases} \quad (1)$$

Given (1), it is easy to verify that the mechanism $\mathcal{M}$ is unbiased, i.e., $\mathbb{E}(\mathcal{M}(x)) = x, \forall x$. Our goal is to design *selection distribution* in the first step such that the mean absolute error $\mathbb{E}(|\mathcal{M}(X) - X|)$ is minimized. In this paper, we assume bin values $\{B_1, \cdots, B_m\}$ are symmetric unless otherwise stated, i.e., $B_i = -B_{m+1-i}, \forall i \in [m]$.

**Selection distribution.** It determines the probability of selecting one bin on the left (or right) of the input $x$ in the first step of our mechanism. Assume $x \in [B_j, B_{j+1})$, then we will select the left index $l \in \{1, \cdots, j\}$ and the right index $r \in \{j + 1, \cdots, m\}$. Let $L_j$ and $R_j$ be the random variables associated with the left index $l$ and right index $r$, respectively, when input $x \in [B_j, B_{j+1})$. Since the probability mass functions (PMF) of both $L_j$ and $R_j$ depend on the value of $j$, we use the following two functions $q_j, q_{m-j}$ to denote their PMF:

$$\Pr\{L_j = i\} := q_j(i), \qquad\qquad i \in \{1, \cdots, j\}$$
$$\Pr\{R_j = i\} := q_{m-j}(m + 1 - i), \quad i \in \{j + 1, \ldots, m\}$$

Note that $q_1(1) = 1$. See Figure 1 for the illustration.

Since both $q_j(\cdot), j \in \{1, 2, \cdots, m\}$ and $\{B_1, \cdots, B_m\}$ are the parameters of mechanism $\mathcal{M}$, we need to design them carefully to minimize the absolute error while satisfying DP

**Algorithm 1** Proposed quantization mechanism $\mathcal{M}$

---

1: **Input:** bin values $B_1, \cdots, B_m$, input $x \in [B_j, B_{j+1})$, PMF $q_j, q_{m-j}$ of $L_j$ and $R_j$.
2: $l \leftarrow i$ w.p. $q_j(i), i \in \{1, \cdots, j\}$.
3: $r \leftarrow i$ w.p. $q_{m-j}(m+1-i), i \in \{j+1, \cdots, m\}$.
4: $\mathcal{M}(x) \leftarrow B_l$ w.p. $\frac{B_r - x}{B_r - B_l}$, and $B_r$ w.p. $\frac{x - B_l}{B_r - B_l}$.
5: **Output** $\mathcal{M}(x)$

---

constraint. We introduce details of finding these parameters in Section 4. Given $q_j(\cdot)$ and $\{B_1, \cdots, B_m\}$, Algorithm 1 summarizes our mechanism $\mathcal{M}$.

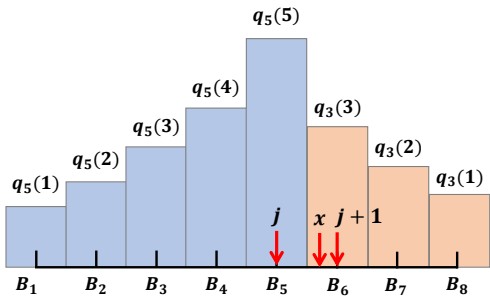

Figure 1: An example of selection distribution

## 3.3 SPECIAL CASES

Section 3.2 presents a general framework for quantization. Indeed, some existing mechanisms proposed in prior works can be regarded as a special case of ours, as detailed below.

**Randomized Quantization Mechanism (RQM) .** It is proposed by Youn et al. [2023] and is a special case of ours. Specifically, $\forall x \in [-c, c]$, RQM randomly outputs one bin from $\{B_1, \cdots, B_m\}$ with

$$B_i = -\Delta - c + (i-1)\frac{2c + 2\Delta}{m - 1}, \ i \in [m],$$

That is, interval $[-c - \Delta, c + \Delta]$ is divided uniformly into $m$ bins. This differs from ours where we enable non-uniformly distributed bins and the bin values are parameters to be optimized (see details in Section 4).

To quantize $x$, RQM first selects a subset of bins: $B_1$ and $B_m$ are selected with probability 1, while among the rest $m - 2$ bins $\{B_2, \cdots, B_{m-1}\}$, each of them is selected independently with probability $q < 1$. Given the selected bins, the one closest to $x$ on the left (resp. right) side is denoted as $B_l$ (resp. $B_r$). Finally, RQM selects either $B_l$ or $B_r$ as the output randomly based on Eq. (1). It turns out that RQM is a special case of our mechanism where selection distribution follows a *Geometric distribution* with parameter $q$, i.e.,

$$q_j(i) = \begin{cases} (1-q)^{j-1}, & \text{if } i = 1 \\ q(1-q)^{j-i}, & \text{if } 1 < i \leq j. \end{cases}$$

**Exponential Randomized Mechanism (ERM).** Inspired by the classic *Exponential Mechanism* (Definition 1), we can propose ERM which outperforms RQM (see the comparison in Section 6) but can still be regarded as a special case of our proposed mechanism. Under ERM, bins are symmetric and satisfy $B_i = -B_{m+1-i}, \forall i \in [m]$. ERM uses a distribution similar to the exponential mechanism for the selection distribution. Specifically, for input $x \in [B_j, B_{j+1})$, PMF $\Pr\{L_j = i\} = q_j(i)$ in ERM depends on the distance between bin $B_i$ and $B_j$ and $\Pr\{R_j = i\} = q_{m-j}(m+1-i)$ depends on the distance between bin $B_i$ and $B_{j+1}$. In other words, ERM uses the following selection distribution:

$$q_j(i) = \frac{\exp\left\{\frac{\gamma(B_i - B_j)}{2(B_j - B_1)}\right\}}{\sum_{k=1}^{j} \exp\left\{\frac{\gamma(B_k - B_j)}{2(B_j - B_1)}\right\}}, \tag{2}$$

where $\gamma$ is a hyperparameter impacting both the privacy and accuracy of $\mathcal{M}$. After obtaining the realizations of $L_j$ and $R_j$, ERM uses Eq. (1)to determine the final output.

Next, we provide privacy and accuracy analysis for ERM. Theorem 1 below provides an upper bound for privacy loss.

**Theorem 1 (Privacy loss of ERM)** *Assume the interval $[-c - \Delta, c + \Delta]$ is divided uniformly into $m$ bins, i.e.,*

$$B_i = -\Delta - c + (i-1)\frac{2c + 2\Delta}{m - 1}, \ i \in [m].$$

*Then ERM satisfies DP with privacy loss*

$$\epsilon < \gamma + \log \frac{2m(c + \Delta)}{c}. \tag{3}$$

The upper bound (3) implies that the privacy loss is an increasing function in the number of bins $m$ and parameter $\gamma$. It is worth noting that according to [Youn et al., 2023], the privacy loss of RQM is bounded by

$$\log\left(\frac{2(1-q)^2(c+\Delta)}{\Delta}\right) + m \log \frac{1}{1-q}.$$

This shows that the privacy loss under RQM also increases in $m$ at the rate of $\mathcal{O}(m)$. In contrast, our ERM has a better privacy loss that increases in $m$ at the rate of $\mathcal{O}(\log m)$.

The next theorem provides an upper bound for the expected absolute error of ERM.

**Theorem 2 (Error of ERM)** *Under the same bins as Theorem 1, the expected absolute error of ERM is bounded:*

$$\mathbb{E}\left(|\mathcal{M}(x) - x|\right) \leq \frac{4}{\gamma} \log(m)(c + \Delta) + \frac{2c + 2\Delta}{m - 1}.$$

The bound implies that when the extended range $\Delta$ increases or the privacy budget parameter $\gamma$ decreases (stricter privacy protection), the performance loss will also increase.

# 4 OPTIMAL MECHANISM

Section 3.2 introduced the general framework of our quantization mechanism. With different bin values $\{B_1, \cdots, B_m\}$ and selection distributions $q_j, q_{m-j}$, we will end up with different mechanisms and we discussed two special cases in Section 3.3. In this section, we explore how to find the optimal mechanism by tuning these parameters. We call the quantization mechanism under the optimal parameter configuration "**OPT**imal randomized quantization **M**echanism (OPTM)." Before introducing OPTM, we first quantify privacy loss and mean absolute error of our mechanism under a given bin values $\{B_1, \cdots, B_m\}$ and selection distributions.

## 4.1 PERFORMANCE MEASURE

Given bin values $\{B_1, \cdots, B_m\}$ and selection distributions $q_j, q_{m-j}$, we can find the *output distribution* $\Pr\{\mathcal{M}(x) = i\}, i \in [m]$ for any input $x$. Let $p(x, i) := \Pr\{\mathcal{M}(x) = i\}$ be the probability that the output of the mechanism $\mathcal{M}$ for an input $x$ is $B_i$. Then, the probability that $\mathcal{M}$ outputs bin $B_l$ on the left of $x$ can be calculated by the law of total probability as follows,

$$p(x, l) = \Pr\{L_j = l\} \sum_{m \geq r \geq j+1} \left( \Pr\{R_j = r\} \frac{B_r - x}{B_r - B_l} \right).$$

Similarly, for a bin $B_r$ on the right side of $x$, we have

$$p(x, r) = \Pr\{R_j = r\} \sum_{1 \leq l \leq j} \left( \Pr\{L_j = l\} \frac{x - B_l}{B_r - B_l} \right).$$

Hence, the output probability of each bin $B_i$ is given by:

$$p(x, i) = \begin{cases} q_j(i) \sum_{r \in [j+1, m]} \left( q_{m-j}(m - r + 1) \frac{B_r - x}{B_r - B_i} \right), & \text{if } B_i \leq x \\ q_{m-j}(m + 1 - i) \sum_{l \in [1, j]} \left( q_j(l) \frac{x - B_l}{B_i - B_l} \right), & \text{o.w.} \end{cases} \quad (4)$$

**Performance measure.** With the output distribution computed above, we can quantify the mean absolute error (MAE) of a mechanism $\mathcal{M}$ as follows,

$$\mathbb{E}\left(|\mathcal{M}(X) - X|\right) = \mathbb{E}_X \left( \sum_{i \in [m]} p(X, i) |B_i - X| \right). \quad (5)$$

To satisfy differential privacy, the output distribution with bounded privacy loss $\epsilon$ should satisfy:

$$\frac{p(x, i)}{p(x', i)} \leq e^\epsilon, \quad \forall x, x' \in [-c, c], i \in [m]. \quad (6)$$

Our goal is to design parameters of $\mathcal{M}$, including $\Delta$, bin values $\{B_1, \cdots, B_m\}$ and especially selection distributions $q_j, q_{m-j}$, such that MAE is minimized subject to bounded privacy loss $\epsilon$. Note that since $m$ determines the number of bits for quantizing $x$ (e.g., 2 bits equals $m = 4$), we assume $m$ is pre-defined and is not a variable to be optimized.

## 4.2 OPTM AS A LINEAR PROGRAM

The problem of finding the optimal parameters of $\mathcal{M}$ can be formulated as an optimization. Our goal is to simplify the optimization as a *linear program* that can be efficiently solved using linear programming tools. Next, we first derive a linear upper bound of the objective function (5). Then, we describe how to turn DP constraint (6) into linear constraints. Finally, we show how to reduce the complexity when the number of output bits is large with another set of constraints.

**Linear upper bound for MAE.** Eq. (5) shows that the mean absolute error is a non-linear function of $q_j(i)$. However, we can find a linear upper bound of it and use it as a proxy, as detailed below.

**Lemma 1** *For any input $x \in [B_j, B_{j+1})$, we have,*

$$\mathbb{E}\left(|\mathcal{M}(x) - x|\right) \leq \frac{1}{2} \left( \zeta_{m-j} + (B_{j+1} - B_j) + \zeta_j \right),$$

*where $\zeta_n = \sum_{i \in [n]} q_n(i)(B_n - B_i)$.*

If we know the distribution of $X$, we can use Lemma 1 to further find a linear upper bound of $\mathbb{E}(|\mathcal{M}(X) - X|)$. An example for uniformly distributed $X$ is given in Theorem 3.

**Theorem 3** *Suppose input $x \in [-c, c]$ follows uniform distribution, $\mathbb{E}(|\mathcal{M}(X) - X|)$ can be upper bounded by*

$$\min_{q_j(i)} \sum_{s \leq n \leq t+1} \left( \min(c, B_n) - \max(-c, B_{n-1}) \right) \left( \zeta_{n-1} + \zeta_{m-n+1} \right), \quad (7)$$

*where $\zeta_n = \sum_{i \in [n]} q_n(i)(B_n - B_i)$. $B_{s-1} \in [-c - \Delta, -c]$ and $B_{t+1} \in (c, c + \Delta]$ are two bins fall in extended range, $B_s < B_t$ are bins in $[-c, c]$ closest to $-c$ and $c$, respectively.*

For more general cases with partially known, non-uniformly, and even asymmetric distributed input $X$, our mechanism can still be adapted.

Specifically, we first change the original definition of selection distribution in Section 3.2 to the following:

$$\Pr\{L_j = i\} := q_j^{(l)}(i), \qquad\qquad i \in \{1, \cdots, j\}$$
$$\Pr\{R_j = i\} := q_{m-j}^{(r)}(m + 1 - i), \quad i \in \{j + 1, \ldots, m\}$$

Both $q_j^{(l)}(\cdot), q_{m-j}^{(r)}(\cdot)$ for all possible $j \in [m]$ are parameters that need to be tuned. Then we can derive a linear upper bound of the mean absolute error (MAE) by extending Lemma 1 and Theorem 3. Theorem 4 below shows the result for non-uniformly distributed $X$ and asymmetric bins.

**Theorem 4** *Suppose input $x \in [-c, c]$ follows any distribution, $\mathbb{E}(|\mathcal{M}(X) - X|)$ can be upper bounded by*

$$\min_{q_j^{(l)}(i), q_{m-j}^{(r)}(i)} \sum_{i=s-1}^{t} (\zeta_{m-i}^{(r)} + B_{i+1} - B_i + \zeta_i^{(l)}) \int_{\max(B_i, -c)}^{\min(B_{i+1}, c)} f_X(x) dx, \quad (8)$$

where $\zeta_{m-j}^{(r)} = \sum_{i \in \{j+1,...,m\}} q_{m-j}^{(r)}(m-i+1)(B_i - B_{j+1})$, $\zeta_j^{(l)} = \sum_{i \in [j]} q_j^{(l)}(i)(B_j - B_i)$. $B_{s-1} \in [-c - \Delta, -c)$ and $B_{t+1} \in (c, c + \Delta]$ are two bins fall in extended range, $B_s < B_t$ are bins in $[-c, c]$ closest to $-c$ and $c$, respectively. $f_X(x)$ is the probability density function of $X$.

Note that the upper bound in Theorem 4 only depends on density $f_X(x)$ through the integral $\Pr(B_i \leq X < B_{i+1}) = \int_{B_i}^{B_{i+1}} f_X(x)dx$, which is easier to know (compared to density itself) and can be estimated from samples.

**Linear differential privacy constraint.** To satisfy $\epsilon$-DP, constraint (6) can be equivalently written as

$$\frac{\max_x p(x, i)}{\min_{x'} p(x', i)} \leq e^\epsilon, \quad \forall i \quad (9)$$

However, constraint (9) is non-linear and we need to convert it to a linear constraint. To this end, we will first show in Lemma 2 that for each $i \in [m]$ and $x \in [-c, c]$, both $\max_x p(x, i)$ and $\min_x p(x, i)$ can be found in a finite set. Such property will then be leveraged to turn constraint (9) into a set of linear constraints.

**Lemma 2** *Assume that* $\forall i, j \in [m], j \geq i$: $q_i(i) \geq q_j(i)$, *then for all input* $x \in [-c, c]$ *and each* $i \in [m]$:

$$\max_x p(x, i) \in \overline{S}_i \quad and \quad \min_x p(x, i) \in \underline{S}_i,$$

*where both* $\overline{S}_i$ *and* $\underline{S}_i$ *are finite sets defined below.*

$$\overline{S}_i = \begin{cases} \{q_i(i), q_{m+1-i}(m+1-i)\}, & \text{if } B_i \in [-c, c] \\ \{p(-c, i)\} \cup \{p(B_k, i) | -c \leq B_k \leq c\}, & \text{if } B_i < -c. \\ \{p(c, i)\} \cup \{p(B_k, i) | -c \leq B_k \leq c\}, & \text{if } B_i > c. \end{cases}$$

$$\underline{S}_i = \begin{cases} \{p(-c, i), p(c, i)\} \cup \left\{ \lim_{x \to B_k} p(x, i) | -c \leq B_k \leq c \right\}, & \text{if } B_i \in [-c, c] \\ \{p(c, i)\} \cup \left\{ \lim_{x \to B_k} p(x, i) | -c \leq B_k \leq c \right\}, & \text{if } B_i < -c. \\ \{p(-c, i)\} \cup \left\{ \lim_{x \to B_k} p(x, i) | -c \leq B_k \leq c \right\}, & \text{if } B_i > c. \end{cases}$$

*where* $\lim_{x \to B_k} p(x, i)$ *above is calculated as follows*

$$\lim_{x \to B_k} p(x, i) = \begin{cases} q_{k-1}(i) \sum_{r \in [k+1,m]} \left( q_{m-k+1}(m-r+1)\frac{B_r - B_k}{B_r - B_i} \right), & \text{if } B_i < B_k. \\ q_{m-k}(m+1-i) \sum_{l \in [1,k-1]} \left( q_k(l)\frac{B_k - B_l}{B_i - B_l} \right), & \text{if } B_i > B_k. \end{cases}$$

Note that $p(x, i)$ is discontinuous and $\lim_{x \to B_k} p(x, i)$ may not equal to $p(B_k, i)$. Lemma 2 shows that for each $i \in [m]$, there is only a finite number of possible values for both $\max_x p(x, i)$ and $\min_x p(x, i)$. Therefore, if we can ensure $\overline{s} \leq e^\epsilon \cdot \underline{s}$ holds for any $\overline{s} \in \overline{S}_i$ and $\underline{s} \in \underline{S}_i$, then privacy constraint (9) is also guaranteed to hold. The monotonicity of the output probability between each pair of bins ensures that we can find a finite set of maximal and minimal probabilities. Example 1 uses specific output distributions of ERM to illustrate this.

**Example 1** *Figure 2 shows two probabilities* $p(x, 3)$ *and* $p(x, 6)$ *of* ERM *when* $m = 8$. *Note that* $\lim_{x \to B_3^+} p(x, 3) = p(B_3, 3) = q_3(3)$ *and* $\lim_{x \to B_3^-} p(x, 3) = q_6(6)$. *We have* $\max_x p(x, i) \in \{q_3(3), q_6(6)\}$. *When* $x$ *increases from* $B_3$ *to* $B_4$, *or decreases from* $B_3$ *to* $-c$, $p(x, 3)$ *decreases. When* $x$ *increases from* $B_4$ *to* $B_5$, $B_5$ *to* $B_6$, $B_6$ *to* $c$, $p(x, 3)$ *also decreases. Hence, we have* $\min_x p(x, 3) \in \{p(-c, 3), \lim_{x \to B_4} p(x, 3), \lim_{x \to B_5} p(x, 3), \lim_{x \to B_6} p(x, 3), p(c, 3)\}$. *The curve of* $p(x, 3)$ *is symmetric to* $p(x, 6)$ *around 0. In Theorem 5, we will use this property to get compact privacy constraints.*

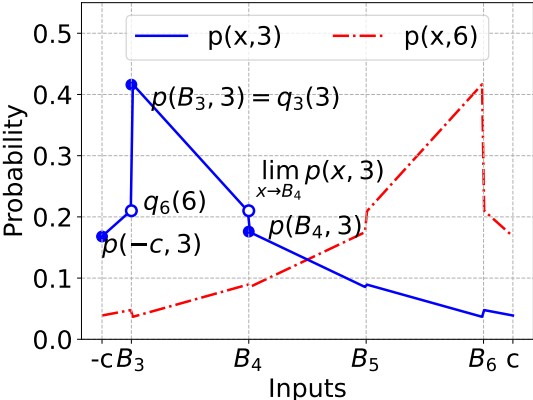

Figure 2: An example of output distribution

However, $\lim_{x \to B_k} p(x, i)$ and $p(x, i)$ are quadratic functions in $q_j(i)$ (see Section 4.1). We still need to convert them into linear forms. To this end, we further assume that each probability $q_j(i)$ has a non-zero lower and upper bound, i.e.,

$$o_j(i) \leq q_j(i) \leq u_j(i), \quad i, j \in [m], i \leq j,$$

where $o_j(i)$ and $u_j(i)$ are hyperparameters that can be found by a grid search. Then, we can replace $q_j(i)$ with $o_j(i)$ or $u_j(i)$ to get a lower bound $w(x, i)$ or upper bound $z(x, i)$ of $p(x, i)$. We illustrate this using an example.

**Example 2** *Consider the following constraint*

$$p(B_k, i) \leq e^\epsilon \cdot \lim_{x \to B_k} p(x, i). \quad (10)$$

*where* $B_i < B_k$, $p(B_k, i) \in \overline{S}_i$ *and* $\lim_{x \to B_k} p(x, i) \in \underline{S}_i$ *by Lemma 2. To make constraint (10) linear, we can replace* $p(B_k, i)$ *with upper bound* $z(B_k, i)$ *and replace* $\lim_{x \to B_k} p(x, i)$ *with a lower bound* $w(B_k, i)$. *Specifically,* $\forall x \in [B_j, B_{j+1})$,

$$z(x, i) = u_j(i) \sum_{r \in [j+1, m]} \left( q_{m-j}(m-r+1)\frac{B_r - x}{B_r - B_i} \right) \quad (11)$$

$$w(x, i) = \begin{cases} o_j(i) \sum_{r \in [j+1, m]} \left( q_{m-j}(m-r+1)\frac{B_r - x}{B_r - B_i} \right), & \text{if } x = -c \text{ or } c. \\ o_{j-1}(i) \sum_{r \in [j+1, m]} \left( q_{m-j+1}(m-r+1)\frac{B_r - B_j}{B_r - B_i} \right), & \text{o.w.} \end{cases}$$

$$(12)$$

*Instead of using constraint* (10)*, we use a stricter version*

$$z(B_k, i) \leq e^\epsilon \cdot w(B_k, i).$$

Similar to Example 2, for any $\overline{s} \in \overline{\mathcal{S}}_i$ and $\underline{s} \in \mathcal{S}_i$, we can turn non-linear constraint $\overline{s} \leq e^\epsilon \cdot \underline{s}$ into a stricter version that is linear. Besides, since the bins and the output distribution are symmetric around 0, we only need to find $\min_x p(x, i)$ from $x \in [B_i, c]$ instead of $[-c, c]$. This results in the constraints detailed in Theorem 5.

**Theorem 5** *If the bins are symmetric, i.e.,* $B_i = -B_{m+1-i}$*, then privacy constraint* (9) *can be satisfied if the following* $\mathcal{O}(m^3)$ *linear constraints are satisfied.*

- $\forall i, k \in [m], -c \leq B_i < B_k \leq c$ :

$$q_i(i) \leq e^\epsilon \cdot w(B_k, i); \quad q_{m+1-j}(m+1-j) \leq e^\epsilon \cdot w(B_k, i);$$
$$q_i(i) \leq e^\epsilon \cdot w(c, i); \quad q_{m+1-j}(m+1-j) \leq e^\epsilon \cdot w(c, i);$$

- $\forall i, k \in [m], B_i < -c < B_k$ :

$$z(B_k, i) \leq e^\epsilon \cdot w(B_k, i); \quad z(B_k, i) \leq e^\epsilon \cdot w(c, i);$$
$$z(-c, i) \leq e^\epsilon \cdot w(B_k, i); \quad z(-c, i) \leq e^\epsilon \cdot w(c, i);$$

- $\forall i, j \in [m], i \leq j, B_j \leq c, B_{j+1} > -c$ :

$$o_j(i) \leq q_j(i) \leq u_j(i); \quad q_i(i) \geq q_j(i)$$

*where* $w(\cdot, \cdot)$ *and* $z(\cdot, \cdot)$ *are specified in* (12) *and* (11).

**Complete optimization.** Combining the above results, we can formulate a linear program for the optimal mechanism. Specifically, we minimize the upper bound of MAE in Theorem 3 subject to 1) a set of linear constraints in Theorem 5, and 2) constraint for distribution $q_j$, i.e., $0 \leq q_j(i) \leq 1, \sum_{i=1}^{j} q_j(i) = 1, \forall i, j$.

The complete procedure for finding the optimal mechanism is shown in Algorithm 2. This optimization can be solved by a linear programming tool denoted by $\mathrm{LinProg}()$, which takes the bin values $\{B_i\}_{i \in [m]}$, privacy parameter $\epsilon$, lower and upper bounds $o_j(i), u_j(i), i, j \in [m], i \leq j$ as inputs and returns the optimal selection distribution $q_j(i)$.

Here we regard $o_j(i), u_j(i)$ as hyperparameters and use grid search to find the optimal ones. Although bin values $\{B_i\}$ are treated as inputs in Algorithm 2, we can find the optimal bins $\{B_i\}$ using techniques such as grid search to further minimize MAE under a fixed privacy parameter $\epsilon$.

**Reduce complexity.** As the number of bins $m$ increases, both the number of $q_j(i)$ and the choice of lower and upper bounds $o_j(i), u_j(i)$ increase. Since the optimal $o_j(i), u_j(i)$ are found via grid search, running Algorithm 2 can be computationally expensive when $m$ is large. Nonetheless, we can formulate the original privacy constraint (9) as another set of linear constraints, which are also stricter but significantly reduce the number of $o_j(i)$ and $u_j(i)$ required to conduct the grid search compared to constraints in Theorem 5.

---

**Algorithm 2** OPTM: find optimal selection distribution

1: **Input:** bin values $\{B_1, \cdots, B_m\}$, privacy parameter $\epsilon$
2: $min\_value = \infty$;
3: $P = \emptyset$;
4: **for** all possible $o_j(i), u_j(i)$ pairs in grid search **do**
5: $\quad obj, \{q_j(i)\} \leftarrow \mathrm{LinProg}\left(\{B_i\}_{i \in [m]}, \epsilon, o_j(i), u_j(i)\right)$;
6: $\quad$ **if** $obj \leq min\_value$ **then**
7: $\quad\quad min\_value \leftarrow obj$;
8: $\quad\quad P \leftarrow \{q_j(i)\}$;
9: $\quad$ **end if**
10: **end for**
11: **Return:** selection probabilities $P$

---

**Theorem 6** *If the bins are symmetric, i.e.,* $B_i = -B_{m+1-i}$*, then the privacy constraint* (9) *can be satisfied if the following linear constraints are satisfied.*

- $\forall i, j \in [m], i \leq j$ :

$$q_j(i) \geq q_{j+1}(i); \quad q_j(i) \leq q_j(i+1);$$

- $\forall i \in [m], -c \leq B(i) \leq c$ :

$$q_i(i) \geq q_{i+1}(i+1);$$

- *Let* $B_s < B_t$ *be bins in* $[-c, c]$ *closest to* $-c$ *and* $c$*, respectively:*

$$z(-c, s-1) \leq e^\epsilon \cdot w(B_t, 1); \quad q_s(s) \leq e^\epsilon \cdot w(B_t, 1);$$
$$z(-c, s-1) \leq e^\epsilon \cdot w(c, 1); \quad q_s(s) \leq e^\epsilon \cdot w(c, 1).$$

- $\forall r, k \in [m], s \leq k \leq t, r > k+1$ :

$$\frac{q_{m-k+1}(m-r+1)}{B_r - B_{k+1}} \geq \frac{q_{m-k}(m-r+1)}{B_r - B_k};$$
$$\frac{q_{m-k}(m-r+1)}{B_r - B_{k+1}} \geq \frac{q_{m-k-1}(m-r+1)}{B_r - B_k};$$

The constraints in Theorem 6 induces that

$$\max_{x,i} p(x, i) \in \{p(-c, s-1), q_s(s)\};$$

$$\min_{x,i} p(x, i) \in \left\{ \lim_{x \to B_t} p(x, 1), p(c, 1) \right\},$$

where $s$ and $t$ are as defined in Theorem 6. Under this set of linear constraints, the upper bound $u_j(i)$ only appears when calculating $z(-c, s-1)$, and the lower bound $o_j(i)$ is only used for computing $w(B_t, 1)$ and $w(c, 1)$. Thus, we can conduct a grid search over 3 variables, regardless of the number of output bits.

## 5   DISCUSSION

Our proposed mechanisms can be generalized to broader settings, including dynamic, high-dimensional, and biased quantization. We discuss these extensions below.

**Extension to high-dimensional quantization.** Besides entry-wise discretization, our method can also be extended to higher-dimensional quantization with a similar method as in [Chaudhuri et al., 2022]. Specifically, for any $d$-dimensional input vector $\mathbf{x} = (\mathbf{x}_1, \cdots, \mathbf{x}_d)$ with $L_2$ norm bounded by diameter $B$, we map the input vector $\mathbf{x}$ to $\mathcal{M}_d(\mathbf{x}) = (\mathcal{M}'(\mathbf{x}_1), \cdots, \mathcal{M}'(\mathbf{x}_d))$. Here, the mechanism $\mathcal{M}'$ quantize the scalar in each coordinate and needs to satisfy $\epsilon$-metric DP, a variant of $\epsilon$-DP that requires the following holds for any two inputs $x, x'$ and any set of possible outputs $S \subseteq \text{Range}(\mathcal{M})$:

$$\Pr(\mathcal{M}'(x) \in S) \leq e^{\epsilon d(x,x')} \Pr(\mathcal{M}'(x') \in S),$$

where $d(x, x') = |x - x'|^2$. Since Lemma 6 in Chaudhuri et al. [2022] has shown that the mechanism $\mathcal{M}_d$ generated by $\epsilon$-metric DP $\mathcal{M}'$ is $\epsilon B^2$-DP and unbiased, we can directly use our method to find the optimal parameters of $\mathcal{M}'$ (under new privacy constraints specified by $\epsilon$-metric DP).

**Extension to biased quantization.** Our unbiased mechanism can be extended to biased quantization, finding a new tradeoff between bias, deviation, and privacy. Instead of randomly outputting either $B_l$ or $B_r$ and enforcing unbiasedness according to Eq. (1) as defined in Section 3.2, we can use the exponential mechanism to output either $B_l$ or $B_r$, with score function being the negative distance between the input and output bins. This mechanism induces biased output but reduces privacy loss.

**Extension to dynamic settings.** Our method can also be extended to dynamic quantization, where different inputs require quantization mechanisms with different hyperparameters. One potential solution is to integrate the existing dynamic quantization strategies, such as the optimal quantization bit-width [Zhou et al., 2018], the clipping range of activation values [So et al., 2024] in a quantized neural network; both methods find hyperparameters (e.g., number of bins, clipping range) during runtime. After these hyperparameters are decided and samples of inputs are collected, we can directly use our algorithm to find the optimal quantization mechanism.

# 6 EXPERIMENTS

Next, we validate two proposed mechanisms: 1) optimal randomized quantization mechanism (OPTM) proposed in Section 4; 2) exponential randomized mechanism (ERM), a special case of OPTM proposed in Section 3.3. We use grid search to find bin values with the best performance.

We conduct three sets of experiments: (i) scalar input quantization; (ii) vector input quantization; and (iii) quantization in stochastic gradient descent (SGD). For each experiment, we compare our mechanisms with two baselines:

- Randomized quantization mechanism (RQM) [Youn et al., 2023]: a special case of OPTM with uniformly-distributed bins as discussed in Section 3.3.
- Minimum variance unbiased (MVU) mechanism [Chaudhuri et al., 2022]: a mechanism that uses optimized probability matrix and output alphabets to map the quantized inputs to outputs. It finds the optimal bin values via a non-linear optimization.

For each mechanism, we evaluate the privacy and accuracy using the standard differential privacy (DP) and mean absolute error (MAE) measures.

## 6.1 SCALAR INPUT

We first evaluate the performance of our algorithm and baselines on a scalar input. In our experiments, the time and resources needed to find the optimal parameters for OPTM are low. It takes about 300 seconds on a personal computer (with Intel Core i5-10210U CPU and 16 GB RAM) to search over all combinations of hyperparameters (10 optional $\Delta$, 10 optional bin assignments, 100 optional lower/upper bounds on probabilities), and find the parameters which can induce the best performance.

We first consider a scenario where input $x$ follows a uniform distribution over $[-1, 1]$. Table 1 compares the mean absolute error $\mathbb{E}_X(|\mathcal{M}(X) - X|)$ at $\epsilon = 0.5, 1.0, 1.5$ when $m = 4$. As expected, our method OPTM improves privacy-accuracy trade-off, and it has the lowest error compared to baselines. The performance of ERM is also comparable with RQM. It is worth mentioning that we could not find valid hyperparameters for RQM and ERM when privacy loss $\epsilon = 0.5$ so we put "N/A" in Table 1. Figure 3 illustrates mean absolute error with higher granularity for each input value $x$. The choice of parameters in each mechanism are given in the Appendix B. We scale the input range of MVU to $[-1, 1]$ for a fair comparison and also scale the output alphabets. The results show that ERM can achieve similar and sometimes better utility than RQM. OPTM can achieve lower error in most cases compared to RQM and ERM, which indicates the effectiveness of the optimization scheme.

We then consider a scenario where input $x$ follows a truncated Gaussian distribution and the distribution is not known in advance. Specifically, the input is first sampled from Gaussian distribution with $\mu = 0.5$, $\sigma = 0.1, 0.2, 0.3$, and then truncated by $[-1, 1]$. Table 2 compares the mean absolute error when $m = 4$ and $\epsilon = 1$. The results show that OPTM can use asymmetric bins to better capture the pattern of the underlying distribution. Specifically, for each optional bin value, we use samples collected from the same input distribution to estimate the density function, optimize for the parameters with the objective function as stated in Theorem 4, and find the bin values inducing the best performance.

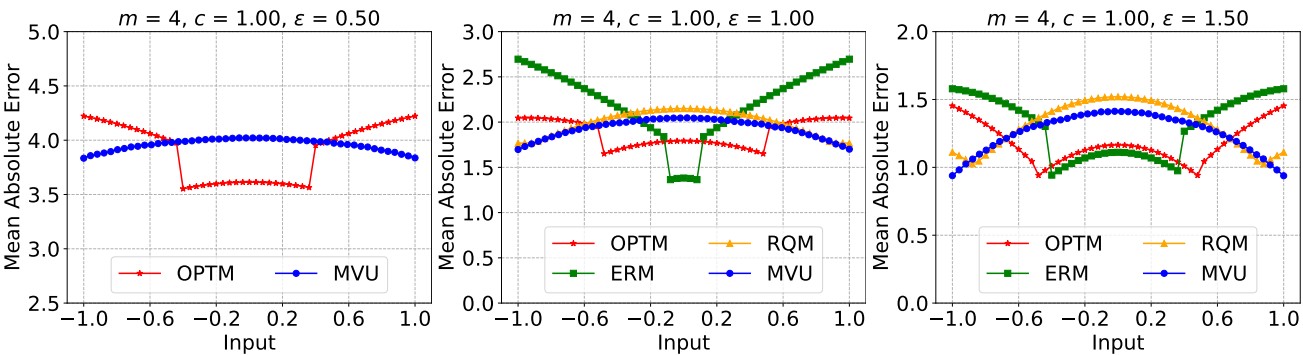

Figure 3: Comparison of mean absolute error under the same privacy on scalar inputs

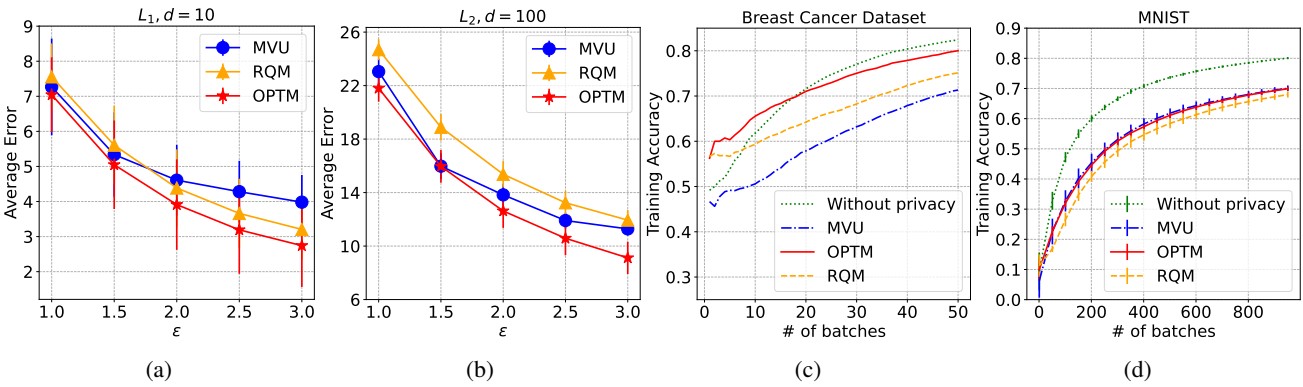

Figure 4: a) Average error of $L_1$ bounded vectors , b) Average error of $L_2$ bounded vectors, c) Training accuracy on breast cancer dataset, d) Training accuracy on MNIST dataset

In comparison, MVU and RQM use uniformly distributed bins for all inputs, hence inducing higher errors. The performance gain brought by asymmetric bins is higher when the distribution is more concentrated (i.e., with smaller $\sigma$).

Table 1: Minimal MAE of scalar inputs under uniform distribution. OPTM attains higher accuracy than baselines. N/A means that there are no valid hyperparameters for ERM and RQM when $\epsilon = 0.5$.

| $\mathbb{E}_X(|\mathcal{M}(X) - X|)$ | $\epsilon = 0.5$ | $\epsilon = 1$ | $\epsilon = 1.5$ |
|---|---|---|---|
| OPTM | 3.904 | 1.882 | 1.179 |
| MVU | 3.959 | 1.930 | 1.254 |
| RQM | N/A | 1.993 | 1.310 |
| ERM | N/A | 2.216 | 1.304 |

## 6.2 VECTOR INPUT

We then compare the error of our mechanism with vector inputs under privacy parameter $\epsilon$. Hyperparameters of each mechanism are given in the Appendix B. We use bounded random vectors as inputs to simulate the clipped gradients in DP-SGD [Abadi et al., 2016], i.e., differentially private stochastic gradient descent commonly used for training pri-

Table 2: Minimal MAE of scalar inputs under truncated Gaussian distribution. Our proposed OPTM attains higher accuracy than baselines.

| $\mathbb{E}_X(|\mathcal{M}(X) - X|)$ | $\sigma = 0.1$ | $\sigma = 0.2$ | $\sigma = 0.3$ |
|---|---|---|---|
| OPTM | 1.778 | 1.836 | 1.972 |
| MVU | 2.053 | 2.052 | 2.002 |
| RQM | 2.028 | 2.010 | 2.000 |

vate machine learning models. Specifically, we generate random vectors with dimension $d = 10$. Each coordinate follows uniform distribution in $[-1, 1]$, hence producing vectors with bounded $L_1$ norm.

For each $\epsilon$, we fix bin values $\{B_1, \ldots, B_m\}$ and find the optimal parameters for each mechanism (e.g., selection probability in OPTM and parameter $q$ for RQM). Then, we quantize each coordinate independently. We measure the Euclidean distance between the input and output vector as the error, and repeat this process 10,000 times to calculate the average error (see Figure 4a).

In another experiment (Figure 4b), we generate random vectors $v \in \mathbb{R}^{100}$ with uniform distribution over ball $||v||_2 \leq 1$ (this can be done through ball point picking [Barthe et al.,

2005]). We quantize the vector $v$ and measure the error based on Euclidean distance. Again we repeat the process 10,000 times to find the average error. We report both the mean and the standard deviation of the error in Figure 4a and 4b. In both cases, OPTM can achieve lower error compared to RQM and MVU, indicating that our mechanism can effectively reduce the loss when privatizing vector inputs.

### 6.3 DP STOCHASTIC GRADIENT DESCENT

We further measure the performance of our mechanisms on downstream machine learning tasks by integrating them into DP-SGD [Abadi et al., 2016] algorithms. Specifically, during each epoch of the Stochastic Gradient Descent (SGD), each coordinate of the gradient vector is clipped by a threshold and then quantized by differentially private mechanisms. The parameters of the experiments are given in the Appendix B. We also record the accuracy when gradients are only clipped, without any privacy protection.

In our experiments, we first use DP-SGD to train a softmax regression model based on the UCI ML Breast Cancer dataset [Wolberg,William, Mangasarian,Olvi, Street,Nick, and Street,W., 1995] with 569 samples. We record the accuracy on the training set after training on each batch of data. Results are shown in Figure 4c. We also train a softmax regression model based on the MNIST dataset [LeCun et al., 2010] with 60,000 images and record the training accuracy. Results are shown in Figure 4d. On the Breast Cancer dataset, OPTM achieves a better convergence rate than RQM and MVU, and achieves very close accuracy compared with the non-private scheme. On MNIST dataset, OPTM has the same performance as MVU and higher accuracy compared to RQM. As errors brought by DP mechanisms can slow down the convergence process, our mechanism can achieve a better convergence rate compared to baselines.

## 7 CONCLUSION

This paper proposes a family of differential privacy mechanisms with discrete and unbiased outputs, which is desirable in many real applications such as federated learning. We design an efficient linear programming algorithm to find the optimal parameters for our mechanism. Experiments on synthetic and real data show that the proposed mechanisms can achieve a better accuracy-privacy trade-off compared with existing discrete differential privacy mechanisms.

Our research raises several interesting topics for future research: (i) Finding the optimal hyperparameters (e.g., number of bins, clipping range) automatically with an optimization during runtime. In this paper, we assume that these hyperparameters are either given in advance or are found using grid search with the best performance. (ii) Privacy analysis of OPTM. This paper only quantifies the privacy

loss of exponential randomized mechanism (ERM) in Theorem 1, finding the privacy loss for the optimal OPTM is important and allows us to better understand the relationship between the privacy bound and mechanism parameters.

## 8 ACKNOWLEDGEMENT

This material is based upon work supported by the U.S. National Science Foundation under award IIS-2202699, IIS-2301599, and ECCS-2301601, by two grants from the Ohio State University Translational Data Analytics Institute, and the College of Engineering Strategic Research Initiative Grant at the Ohio State University.

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

# Privacy-Aware Randomized Quantization via Linear Programming
## (Supplementary Material)

**Zhongteng Cai**[1]  **Xueru Zhang**[1]  **Mohammad Mahdi Khalili**[1,2]

[1]Department of Computer Science and Engineering, Ohio State University, Columbus, Ohio, USA
[2] Yahoo Research, New York, New York, USA

## A PROOFS

**Proof 1** *(of Theorem 1):*

*Assume that bins are uniformly distributed, i.e., $B_i = -\Delta - c + (i-1)\frac{2c+2\Delta}{m-1}$ ($i \in [m]$), $m \geq 4$, the selection probability of ERM can be calculated as:*

$$q_j(i) = \frac{\exp\{\frac{\gamma(B_i - B_j)}{2(B_j - B_1)}\}}{\sum_{k=1}^{j} \exp\{\frac{\gamma(B_k - B_j)}{2(B_j - B_1)}\}} = \frac{\exp\{\frac{\gamma(i-j)}{2(j-1)}\}}{\sum_{k=1}^{j} \exp\{\frac{\gamma(k-j)}{2(j-1)}\}}.$$

*Take reciprocal of the probability, we have:*

$$\frac{1}{q_j(i)} = \sum_{k=1}^{j} \exp\{\frac{\gamma(k-i)}{2(j-1)}\} \leq j \exp\{\frac{\gamma(j-1)}{2(j-1)}\} = j \exp\{\frac{\gamma}{2}\}.$$

*Therefore we can find the lower bound of $q_j(i)$:*

$$q_j(i) \geq \frac{1}{j} \exp\{-\frac{\gamma}{2}\}.$$

*Combining the lower bound with (4), we obtain a lower bound of $p(x,i)$:*

$$p(x,i) = q_j(i) \sum_{m \geq r \geq j+1} \left( q_{m-j}(m-r+1)\frac{B_r - x}{B_r - B_i} \right) \geq \frac{\exp\{-\frac{\gamma}{2}\}}{j} \cdot \frac{\exp\{-\frac{\gamma}{2}\}}{m-j} \cdot \sum_{m \geq r \geq j+1} \left( \frac{B_r - x}{B_r - B_i} \right)$$

$$\geq \frac{\exp\{-\gamma\}}{m^2/4} \cdot \sum_{m \geq r \geq j+1} \left( \frac{B_r - x}{B_r - B_i} \right) \geq \frac{4\exp\{-\gamma\}}{m^2(B_m - B_1)} \cdot \sum_{m \geq r \geq j+1}(B_r - B_{j+1})$$

$$\geq \frac{4\exp\{-\gamma\}}{m^2(2c+2\Delta)} \cdot \frac{2c+2\Delta}{m-1} \cdot \sum_{m \geq r \geq j+1}(r - (j+1)) = \frac{4\exp\{-\gamma\}}{m^2(m-1)} \cdot \frac{(m-j-1)(m-j)}{2}.$$

*Since $x \in [-c, c]$ and $x \in [B_j, B_{j+1})$, we have:*

$$(j-1) \cdot \frac{2c+2\Delta}{m-1} < \Delta \leq j \cdot \frac{2c+2\Delta}{m-1},$$

*which implies:*

$$\frac{(m-1)\Delta}{2c+2\Delta} \leq j < \frac{(m-1)\Delta}{2c+2\Delta} + 1.$$

*Hence we have:*

$$p(x,i) \geq \frac{4\exp\{-\gamma\}}{m^2(m-1)} \cdot \frac{(m-j-1)(m-j)}{2} > \frac{4\exp\{-\gamma\}}{m^2(m-1)} \cdot \frac{(m-\frac{(m-1)\Delta}{2c+2\Delta}-2)(m-\frac{(m-1)\Delta}{2c+2\Delta}-1)}{2}$$

$$= \frac{(2mc+m\Delta-4c-4\Delta)(2mc+m\Delta-2c-2\Delta)\exp\{-\gamma\}}{2m^2(m-1)(c+\Delta)} > \frac{c\exp\{-\gamma\}}{2m(c+\Delta)} \quad (m \geq 4).$$

*Let privacy loss $e^\epsilon = \max_{x,x'} \frac{p(x,i)}{p(x',i)}$, $i \in [m]$, we have:*

$$e^\epsilon \leq \frac{\max_{x,i} p(x,i)}{\min_{x,i} p(x,i)} \leq \frac{1}{\frac{c\exp\{-\gamma\}}{2m(c+\Delta)}} = \frac{2m(c+\Delta)\exp\gamma}{c}.$$

*Hence we have an upper bound on $\epsilon$:*

$$\epsilon \leq \gamma + \log \frac{2m(c+\Delta)}{c}.$$

**Proof 2** *(of Theorem 2):*

*For exponential mechanism with $\delta$-sensitive score function $f$, privacy parameter $\gamma$, set of output $\mathcal{Y}$, we have the following inequality on the quality $f(y)$ of the output $y$ [Bassily et al., 2016]:*

$$\mathbb{E}(f(y)) \geq \max_{y \in \mathcal{Y}} f(y) - \frac{2\delta \log |\mathcal{Y}|}{\gamma}.$$

*Assume $x \in [B_j, B_{j+1})$, $B_i = -c - \Delta + (i-1)\frac{2c+2\Delta}{m-1}$. When selecting the left bin $B_l$ with exponential mechanism (denote as event $L_j = l$), we have $f(l) = B_j - B_l$, $\max f(l) = 0$, sensitivity of the score function $\delta = B_j - B_1$, $|\mathcal{Y}| = j$, hence we have:*

$$\mathbb{E}(B_j - B_l) = -\mathbb{E}(q(l)) \leq \frac{2(B_j - B_1)\log j}{\gamma}.$$

*Similarly, when selecting the right bin $B_r$, we have:*

$$\mathbb{E}(B_r - B_{j+1}) \leq \frac{(B_m - B_{j+1})\log(m-j)}{\gamma}.$$

*Since $\mathcal{M}(x) \in \{B_l, B_r\}$, we can have an upper bound on the expected absolute error:*

$$\mathbb{E}(|\mathcal{M}(x) - x|) = \mathbb{E}(B_r - B_{j+1}) + (B_{j+1} - B_j) + \mathbb{E}(B_j - B_l)$$

$$\leq \frac{2(B_j - B_1)\log j}{\gamma} + \frac{2c+2\Delta}{m-1} + \frac{(B_m - B_{j+1})\log(m-j)}{\gamma}$$

$$\leq \frac{2\log(m)(2c+2\Delta)}{\gamma} + \frac{2c+2\Delta}{m-1}.$$

**Proof 3** *(of Lemma 1):*

*For each given $x \in [B_j, B_{j+1})$, the upper bound of its Mean Absolute Error can be derived as follows:*

$$
\begin{aligned}
\mathbb{E}(|\mathcal{M}(x) - x|) &= \sum_{\substack{i \leq j \\ k > j}} \Pr(L_j = i) \Pr(R_j = k)\Big(\big(\frac{B_k - x}{B_k - B_i}\big)(x - B_i) + \big(\frac{x - B_i}{B_k - B_i}\big)(B_k - x)\Big) \\
&= \sum_{\substack{i \in [1,j] \\ k \in [j+1,m]}} \Pr(L_j = i) \Pr(R_j = k)\big(\frac{2(x - B_i)(B_k - x)}{B_k - B_i}\big) \\
&\leq \sum_{\substack{i \in [1,j] \\ k \in [j+1,m]}} \Pr(L_j = i) \Pr(R_j = k)\big(\frac{B_k - B_i}{2}\big) \\
&= \frac{1}{2}\mathbb{E}(B_r - B_l) \\
&= \frac{1}{2}\big(\mathbb{E}(B_r - B_{j+1}) + \mathbb{E}(B_{j+1} - B_j) + \mathbb{E}(B_j - B_l)\big),
\end{aligned}
$$

*where $B_r$ is the random variable denoting the bin selected on the right, and $B_l$ is the random variable denoting the bin selected on the left.*

*Considering the process of selecting one bin from $n$ bins: $B_1, B_2, \cdots, B_n$ according to the selection probability $q_n(1), q_n(2), \cdots, q_n(n)$. Denote the expected distance between the output bin $B_i$ and $B_n$ as $\zeta_j$. $\zeta_n = \sum_{i \in [1,n]} q_n(i)(B_n - B_i)$, which is the linear combination of selection probabilities when the value of bins are fixed. We know that $\mathbb{E}(B_r - B_{j+1}) = \zeta_{m-j}$, and $\mathbb{E}(B_j - B_l) = \zeta_j$. Hence we obtain:*

$$
\mathbb{E}(|\mathcal{M}(x) - x|) \leq \frac{1}{2}\big(\zeta_{m-j} + (B_{j+1} - B_j) + \zeta_j\big).
$$

**Proof 4** *(of Theorem 3):*

*Assume that the position of bins are given (either uniformly or non-uniformly distributed), and the input $x \in [-c, c]$ follows uniform distribution, and the probability density function of $X$ is equal to $f_X(x) = \frac{1}{2c}$. Then, we find an upper bound for $\mathbb{E}(|\mathcal{M}(X) - X|)$ using Lemma 1 and law of total expectation as follows,*

$$
\mathbb{E}(|\mathcal{M}(X) - X|) = \int_{-c}^{c} \frac{1}{2c}\mathbb{E}(|\mathcal{M}(x) - x|)dx
$$

$$
= \int_{-c}^{B_s} \frac{1}{2c}\mathbb{E}(|\mathcal{M}(x) - x|)dx + \sum_{i=s}^{t-1}\int_{B_i}^{B_{i+1}} \frac{1}{2c}\mathbb{E}(|\mathcal{M}(x) - x|)dx + \int_{B_t}^{c} \frac{1}{2c}\mathbb{E}(|\mathcal{M}(x) - x|)dx
$$

$$
\leq \frac{1}{2c}\Big((B_s + c)(\zeta_{m-s+1} + B_{s+1} - B_s + \zeta_{s-1}) + \sum_{i=s}^{t-1}(B_{i+1} - B_i)(\zeta_{m-i} + B_{i+1} - B_i + \zeta_i) + (c - B_t)(\zeta_{m-t} + B_{t+1} - B_t + \zeta_t)\Big),
$$

*where $-c - \Delta \leq B_{s-1} < -c \leq B_s < B_t \leq c < B_{t+1} \leq c + \Delta$. Discarding the constant terms, we can have the following objective function:*

$$
\min_{q_j(i)} \sum_{s \leq n \leq t+1} \big(\min(c, B_n) - \max(-c, B_{n-1})\big)\big(\zeta_{n-1} + \zeta_{m-n+1}\big), \tag{13}
$$

*where $\zeta_n$ is given in Lemma 1 and Theorem 3.*

**Proof 5** *(of Lemma 2):*

*When $-c \leq B_i \leq c$ and $x \geq B_i$, according to (4), $p(x, i) = q_j(i)\sum_{m \geq r \geq j+1}\Big(q_{m-j}(m - r + 1)\frac{B_r - x}{B_r - B_i}\Big)$. Since $\frac{B_r - x}{B_r - B_i} < 1$, we obtain that:*

$$
p(x, i) \leq q_j(i)\sum_{m \geq r \geq j+1} q_{m-j}(m - r + 1) = q_j(i).
$$

$\forall j \in [m], j \geq i$, we assume $q_i(i) \geq q_j(i)$, hence we have $q_i(i) \in \overline{\mathcal{S}}_i$, where $\overline{\mathcal{S}}_i$ is as defined in Lemma 2. Similarly, we can prove that when $-c \leq B_i \leq c$ and $x < B_i$, $q_{m+1-i}(m+1-i) \in \overline{\mathcal{S}}_i$.

If $B_i \leq x$, then $\forall x, x' \in [B_k, B_{k+1}), x \leq x'$, we have $p(x, i) \geq p(x', i)$. This indicates that $p(x, i)$ is monotonic between each interval divided by bins (e.g., $[-c, B_i)$, $[B_i, B_{i+1})$, or $[B_i, c)$), and is decreasing as $x$ is moving farther from $B_i$. We can also prove this when $x < B_i$. Hence if $B_i < -c$, $\max p(x, i)$ is achieved only when $x = -c$ or $x = B_k$ ($-c \leq B_k < c$). If $B_i > c$, then $\max p(x, i)$ is achieved only when $x = c$ or $x = B_k$ ($-c \leq B_k < c$). Similarly, $\min p(x, i)$ is achieved only when $x$ is approaching the position of bins ($B_k$), or locating at the edge ($c$ or $-c$) which is farther from $B_i$.

**Proof 6** *(of Theorem 6):*

*According to Lemma 2, when $B_i \leq -c$, we have:*

$$\max_x p(x, i) \in \{p(-c, i)\} \cup \{p(B_k, i)\}(k \in [m], -c \leq B_k \leq c).$$

*According to (4), when $B_i \leq x$:*

$$p(x, i) = q_j(i) \sum\nolimits_{m \geq r \geq j+1} \left( q_{m-j}(m - r + 1) \frac{B_r - x}{B_r - B_i} \right).$$

*Hence we can get:*

$$p(B_k, i) = q_k(i) \sum\nolimits_{m \geq r \geq k+1} \left( q_{m-k}(m - r + 1) \frac{B_r - B_k}{B_r - B_i} \right).$$

$$p(B_{k+1}, i) = q_{k+1}(i) \sum\nolimits_{m \geq r \geq k+2} \left( q_{m-k-1}(m - r + 1) \frac{B_r - B_{k+1}}{B_r - B_i} \right).$$

*Assume that $\forall i, j \in [m], i \leq j$:*

$$q_j(i) \geq q_{j+1}(i), \tag{14}$$

*and $\forall k, r \in [m], s \leq k \leq t, r > k + 1$ (s and t are as defined in Theorem 3), we assume:*

$$q_{m-k}(m - r + 1) \cdot (B_r - B_k) \geq q_{m-k-1}(m - r + 1) \cdot (B_r - B_{k+1}),$$

*then we get:*

$$p(B_k, i) \geq p(B_{k+1}, i). \tag{15}$$

*From (4), we can also know that:*

$$p(x, i - 1) = q_j(i - 1) \sum\nolimits_{m \geq r \geq j+1} \left( q_{m-j}(m - r + 1) \frac{B_r - x}{B_r - B_{i-1}} \right).$$

*Assume that $\forall i, j \in [m], i \leq j$, we have:*

$$q_j(i - 1) \leq q_j(i), \tag{16}$$

*hence we can know that:*

$$p(x, i - 1) \leq p(x, i). \tag{17}$$

*Through (15), (17), and Lemma 2, we can know that when $B_i < -c$, we have $p(-c, s-1) \in \overline{S}_i$, where $\overline{S}_i$ is as defined in Lemma 2, $s$ is as defined in Theorem 3.*

*When $-c \leq B_i \leq c$, we have $\max_x p(x, i) \in \{q_i(i), q_{m+1-i}(m+1-i)\}$. Assume that $\forall i \in [m], q_i(i) \geq q_{i+1}(i+1)$, we have $q_s(s) \in \overline{S}_i$. Now we have $\overline{S}_i = \{p(-c, s-1), q_s(s)\}$.*

*According to Lemma 2 and (17), $\min_{x,i} p(x, i) \in \{\lim_{x \to B_k} p(x, 1) | -c \leq B_k \leq c\} \cup \{p(c, 1)\}$. We have:*

$$\lim_{x \to B_k} p(x, 1) = q_{k-1}(1) \sum_{r \in [k+1, m]} \left( q_{m-k+1}(m-r+1) \frac{B_r - B_k}{B_r - B_1} \right), \tag{18}$$

$$\lim_{x \to B_{k+1}} p(x, 1) = q_k(1) \sum_{r \in [k+2, m]} \left( q_{m-k}(m-r+1) \frac{B_r - B_{k+1}}{B_r - B_1} \right). \tag{19}$$

*According to (14), we have $q_{k-1}(j) \geq q_k(j)$, hence by requiring that for any $r, k \in [m], s \leq k \leq t, r > k + 1$:*

$$q_{m-k+1}(m-r+1)(B_r - B_k) \geq q_{m-k}(m-r+1)(B_r - B_{k+1}), \tag{20}$$

*we obtain:*

$$\lim_{x \to B_k} p(x, 1) > \lim_{x \to B_{k+1}} p(x, 1). \tag{21}$$

*Combining (21) with Lemma 2, we can know that:*

$$\min_x p(x, i) \in \{ \lim_{x \to B_t} p(x, 1), p(c, 1) \}. \tag{22}$$

# B EXPERIMENTAL DETAILS

The hyperparameters used in each experiment are given as follows.

| Hyperparameter | Value |
|---|---|
| OPTM bins | [-6.00, -0.40, 0.40, 6.00] |
| MVU bins | [-4.34, -3.60, 3.60, 4.34] |

Table 3: Hyperparameters for scalar inputs when $\epsilon = 0.5$

| Hyperparameter | Value |
|---|---|
| OPTM bins | [-3.00, -0.50, 0.50, 3.00] |
| MVU bins | [-2.42, -1.69, 1.69, 2.42] |
| ERM $\gamma$ | 0.026 |
| ERM bins | [-5.10, -0.10, 0.10, 5.10] |
| RQM $q$ | 0.220 |
| RQM bins | [-2.70, -0.90, 0.90, 2.70] |

Table 4: Hyperparameters for scalar inputs when $\epsilon = 1.0$

| Hyperparameter | Value |
|---|---|
| OPTM bins | [-3.00, -0.50, 0.50, 3.00] |
| MVU bins | [-1.83, -1.11, 1.11, 1.83] |
| ERM $\gamma$ | 0.043 |
| ERM bins | [-2.70, -0.40, 0.40, 2.70] |
| RQM $q$ | 0.498 |
| RQM bins | [-2.60, -0.87, 0.87, 2.60] |

Table 5: Hyperparameters for scalar inputs when $\epsilon = 1.5$

| Hyperparameter | Value |
|---|---|
| OPTM bins | [-4.00, 0.20, 0.60, 4.00] |
| MVU bins | [-2.42, -1.69, 1.69, 2.42] |
| RQM $q$ | 0.220 |
| RQM bins | [-2.70, -0.90, 0.90, 2.70] |

Table 6: Hyperparameters for truncated Gaussian distribution

| Hyperparameter | Value |
|---|---|
| OPTM bins | [-3, -0.5, 0.5, 3] |
| RQM bins | [-3, -1, 1, 3] |

Table 7: Hyperparameters for vector inputs

| Hyperparameter | Value |
|---|---|
| Batch size | 8 |
| DP budget $\epsilon$ | 1 |
| Gradient norm clip | 0.1 |
| OPTM bins | [-2.2, -0.4, 0.4, 2.2] |
| RQM bins | [-2.7, -0.9, 0.9, 2.7] |
| RQM $q$ | 0.22 |

Table 8: Hyperparameters for DP-SGD on Breast Cancer dataset

| Hyperparameter | Value |
|---|---|
| Batch size | 32 |
| DP budget $\epsilon$ | 1 |
| Gradient norm clip | 0.01 |
| OPTM bins | [-2.6, -0.4, 0.4, 2.6] |
| RQM bins | [-2.7, -0.9, 0.9, 2.7] |
| RQM $q$ | 0.22 |

Table 9: Hyperparameters for
DP-SGD on MNIST