# OpenReview forum: "Privacy-Aware Randomized Quantization via Linear Programming"
_auai.org/UAI/2024/Conference — UAI 2024 poster_

### Official Review · Reviewer_csQm · 2024-03-20

**Q2-1 Originality-Novelty:** 3
**Q2-2 Correctness-Technical Quality:** 3
**Q2-5 Clarity Of Writing:** 3

**Q1 Summary And Contributions:**

The paper addresses the problem of discrete differential privacy. It gives a general mechanism that exhibits differential privacy and shows that existing algorithms are a special case of their generalization. They describe techniques to optimize their generalized mechanism using linear programming and provide empirical evidence for showing advantages of their approach.

**Q2-3 Extent To Which Claims Are Supported By Evidence:**

3: Good: the main claims are supported by convincing evidence (in the form of adequate experimental evaluation, proofs, (pseudo-)code, references, assumptions).

**Q2-4 Reproducibility:**

3: Good: key resources (e.g. proofs, code, data) are available and key details (e.g. proofs, experimental setup) are sufficiently well-described for competent researchers to confidently reproduce the main results.

**Q3 Main Strengths:**

1. The writing and the presentation of the paper is very clear.
2. The generalization in the paper have already existing special cases which shows the utility of the mathematical setup
3. Th experimental setup seems sufficient for evidence on privacy-accuracy tradeoff.

**Q4 Main Weakness:**

1. The experimental evidence does not talk about time and resources needed in finding the optimal parameters for OPTM
2. The bins were hyperparameters fixed for each experiment, it would be good if authors can provide some intuition as to how to choose these bins.
3. Section 4.2 needs a better presentation. A lot of mathematical formulas with little intuition are a bit hard to follow.

**Q5 Detailed Comments To The Authors:**

1. Can authors please explain the choice of symmetric bins. I believe it was to make the linear optimization problem easier. Is that the case?
2. I would imagine as m, the number of bins increase, the privacy guarantees would converge to that of the continuous case. If that is the case, I believe it would add more value to the overall setup.

**Q9 Complying With Reviewing Instructions:**

Yes

---

> ### Author Rebuttal · Authors · 2024-04-09
>
> Thanks for your comments. Here are our responses point by point:
>
> > Time and resources
>
> It takes about 300 seconds on a personal computer (with Intel Core i5-10210U CPU and 16 GB RAM) to search over all combinations of hyperparameters (10 optional $\delta$, 10 optional bin assignments, 100 optional lower/upper bounds on probabilities), optimize the parameters for each combination, and find the parameters which can induce the best performance.
>
> > Bins in experiment
>
> We will add more explanation about how we choose bins in the revised manuscript. RQM uses uniformly-distributed bins [R1]. MVU uses non-linear optimization to find the optimal bin values [R2]. For ERM and OPTM, which are proposed in this paper and enable non-uniformly distributed bins, we use grid search to find bin values with optimal performance.
>
> > Intuition about the contents of Section 4.2
>
> We will add more intuitive explanations in the revised manuscript. Here is a brief explanation of Section 4.2. First, we can derive an upper bound on $\mathbb{E}(|\mathcal{M}(X)-X|)$ as the optimization objective, which is a linear function of selection distribution parameters (Theorem 3). Next, we derive linear constraints to enforce differential privacy (Theorem 4). For each bin index $i \in [m]$, $\max_x \mathcal{M}(x)$ and $\min_x \mathcal{M}(x)$ can be found in finite sets $\overline{\mathcal{S}}_i$ and $\underline{\mathcal{S}}_i$, respectively (Lemma 2). For any $\overline{s} \in \overline{\mathcal{S}}_i$ and $\underline{s} \in \underline{\mathcal{S}}_i$, we enforce $\overline{s} \leq e^{\epsilon} \cdot \underline{s}$. Some of the elements in $\overline{\mathcal{S}}_i$ and $\underline{\mathcal{S}}_i$ are quadratic functions in parameters we want to optimize, but we can convert them into linear terms by replacing some parameters with their lower or upper bounds, which are hyperparameters that can be found by a grid search. Therefore, we have a complete optimization problem that can be solved by linear programming.
>
> > The reason of choosing symmetric bins
>
> The main reason why we choose symmetric bins is that we assume input $X$ follows uniform distribution, hence assigning asymmetric bins will not provide additional performance gain. Symmetric bins can also reduce the number of parameters we need to optimize. However, we emphasize that our method can be extended to asymmetric bins when the distribution of $X$ is not uniform and is only (partially) known in advance.
> A detailed explanation of how to extend our mechanism is as follows.
>
> First, we change the original definition of selection distribution in Section 3.2 to the following:
> $$\Pr\\{L_j = i\\}:=q_j^{(l)}(i),i\in \\{1,\cdots,j\\};$$
> $$\Pr\\{R_j = i\\}:=q_{m-j}^{(r)}(m+1-i),i \in \\{j+1,\ldots,m\\}.$$
>
> We need to tune both $q_j^{(l)}(\cdot), q_{m-j}^{(r)}(\cdot)$ for all possible $j \in [m]$.
>
> Second, we derive an upper bound of the mean absolute error (MAE) of the mechanism, which is a linear function of parameters $q_j^{(l)}(\cdot), q_{m-j}^{(r)}(\cdot)$ (this can be proved by directly extending Lemma 1 and Theorem 3). We can find a linear upper bound on the expectation of MAE as follows:
> $$\mathbb{E}(|\mathcal{M}(X)-X|)
>     \leq \sum_{i=s-1}^{t}  (\zeta_{m-i}^{(r)}+B_{i+1}-B_{i}+\zeta_{i}^{(l)}) \int_{\max(B_i, -c)}^{\min(B_{i+1}, c)} f_X(x) dx,$$
> where $f_X(x)$ is the probability density function of $X$; $B_s<B_t$ are bins in $[-c,c]$ closest to $-c$ and $c$, respectively; $\zeta_j^{(l)} = \sum_{i \in [j]} q_j^{(l)}(i) (B_{j}-B_{i})$, $\zeta_{m-j}^{(r)} = \sum_{i \in \\{j+1, \dots, m\\}} q_{m-j}^{(r)}(m-i+1) (B_{i}-B_{j+1})$. Note that this upper bound only depends on density $f_X(x)$ through the integral $\Pr(B_i \leq X < B_{i+1}) = \int_{B_i}^{B_{i+1}} f_X(x)dx$, which is easier to know (compared to density itself) and can be estimated from samples.
>
> Finally, we can minimize this upper bound with privacy constraints through linear programming, similar to the methods stated in Lemma 2, Theorem 4, and Algorithm 2.
>
> > The privacy guarantee when $m$ increases
>
> When $m$ increases to infinity, our mechanism will be equivalent to the following mechanism: generates a truncated positive noise $y_1$ and a truncated negative noise $y_2$ for the input $x$, then randomly outputs either $x+y_1$ or $x+y_2$ with unbiased expectation. Such a mechanism is different from any existing continuous differential privacy mechanisms. Quantifying the privacy loss of this mechanism is challenging. Analysing whether it can outperform existing mechanisms such as Laplace or Gaussian mechanism will be an interesting future research topic.
>
> [R1] Youn, Yeojoon, et al. "Randomized quantization is all you need for differential privacy in federated learning." arXiv preprint arXiv:2306.11913 (2023).
>
> [R2] Chaudhuri, Kamalika, Chuan Guo, and Mike Rabbat. "Privacy-aware compression for federated data analysis." Uncertainty in Artificial Intelligence. PMLR, 2022.

---

### Official Review · Reviewer_Wvpr · 2024-03-22

**Q2-1 Originality-Novelty:** 2
**Q2-2 Correctness-Technical Quality:** 2
**Q2-5 Clarity Of Writing:** 3

**Q1 Summary And Contributions:**

The paper introduces a new family of differentially private quantization mechanisms, extending the Randomized Quantization Mechanism (RQM), to produce discrete and unbiased outputs. Theoretically, they demonstrate that the expected absolute error of ERM, a specific variant of their proposed mechanisms, scales better with the number of bins in quantization $m$. The expected absolute error of ERM increases with $m$ at the rate $O(\log (m))$, while that of RQM increases at the rate $O(m)$.


For uniformly distributed data, the authors further design a linear program to find the mechanism with smallest expected absolute error within the family of DP quantization mechanisms. Empirical results support their claims when the number of bins is small $m = 4$.

**Q2-3 Extent To Which Claims Are Supported By Evidence:**

2: Fair: the main claims are somewhat supported by evidence (but the experimental evaluation may be weak, or does not match entirely with the claims, important baselines may be missing, proofs contain important ideas but lack rigor, algorithmic details are only discussed superficially, references are imprecise, assumptions are not sufficiently motivated or explicated, etc.).

**Q2-4 Reproducibility:**

3: Good: key resources (e.g. proofs, code, data) are available and key details (e.g. proofs, experimental setup) are sufficiently well-described for competent researchers to confidently reproduce the main results.

**Q3 Main Strengths:**

1. The paper proposed a flexible framework for unbiased quantization with differential privacy guarantees. Particularly, for a specific variant (ERM), their theoretical results offer insights into the conditions under which ERM outperforms other methods.

2. Their experimental results corroborate their claims on the performance of quantization with linear programming, when the data follows uniform distributions (Table 1). Also, their method can generalize effectively to higher-dimensional settings (d = 100) from experiments in Figure 4b.

3. The paper is clearly written.

**Q4 Main Weakness:**

1. Theorem 3 states a linear upper bound when the data is uniformly distributed. However, real world data usually deviates from uniform distributions. Adapting the proposed method to non-uniformly distributed data is case-by-case and non-trivial, therefore limiting the application of the proposed linear programming method.

2. It seems to me that the designing of selection distribution requires knowledge of the parameter j, an index of the bin value satisfying $B_j \leq x \leq B_{j+1}$. The value $j$ contains information about a data point $x$. Can you provide an intuitive explanation on why this does not contribute to additional privacy loss?

**Q5 Detailed Comments To The Authors:**

Questions:
1. In Theorem 2, what is the distribution of X? Does it hold for any distribution of X?
2. For the experiment in table 2, ERM seems to perform badly when $m$ is small. Are there results for larger $m$?

**Q9 Complying With Reviewing Instructions:**

Yes

---

> ### Author Rebuttal · Authors · 2024-04-09
>
> Thanks for your comments. Here are our responses point by point:
>
> > Non-uniformly distributed inputs
>
> In Theorem 3, we assume $X$ follows uniform distribution. But our mechanism can be generalized to non-uniform or partially known distribution of input $X$, and we can assign non-uniformly distributed and asymmetric bins to capture the pattern of the distribution. Specifically, when input distribution is unknown, we can first collect samples of inputs. For any given bin values, we can estimate the probability that inputs fall into the intervals between different pairs of neighboring bins (e.g., $\Pr\\{B_i \leq X < B_{i+1}\\}$). Then we can derive an optimization objective based on this estimation and optimize the parameters accordingly. A detailed explanation is as follows.
>
> First, we change the original definition of selection distribution in Section 3.2 to the following:
> $$\Pr\\{L_j = i\\}:=q_j^{(l)}(i),i\in \\{1,\cdots,j\\};$$
> $$\Pr\\{R_j = i\\}:=q_{m-j}^{(r)}(m+1-i),i \in \\{j+1,\ldots,m\\}.$$
>
> We need to tune both $q_j^{(l)}(\cdot), q_{m-j}^{(r)}(\cdot)$ for all possible $j \in [m]$.
>
> Second, we derive an upper bound of the mean absolute error (MAE) of the mechanism, which is a linear function of parameters $q_j^{(l)}(\cdot), q_{m-j}^{(r)}(\cdot)$ (this can be proved by directly extending Lemma 1 and Theorem 3). We can find a linear upper bound on the expectation of MAE as follows:
> $$\mathbb{E}(|\mathcal{M}(X)-X|)
>     \leq \sum_{i=s-1}^{t}  (\zeta_{m-i}^{(r)}+B_{i+1}-B_{i}+\zeta_{i}^{(l)}) \int_{\max(B_i, -c)}^{\min(B_{i+1}, c)} f_X(x) dx,$$
> where $f_X(x)$ is the probability density function of $X$; $B_s<B_t$ are bins in $[-c,c]$ closest to $-c$ and $c$, respectively; $\zeta_j^{(l)} = \sum_{i \in [j]} q_j^{(l)}(i) (B_{j}-B_{i})$, $\zeta_{m-j}^{(r)} = \sum_{i \in \\{j+1, \dots, m\\}} q_{m-j}^{(r)}(m-i+1) (B_{i}-B_{j+1})$. Note that this upper bound only depends on density $f_X(x)$ through the integral $\Pr(B_i \leq X < B_{i+1}) = \int_{B_i}^{B_{i+1}} f_X(x)dx$, which is easier to know (compared to density itself) and can be estimated from samples.
>
> Finally, we can minimize this upper bound with privacy constraints through linear programming, similar to the methods stated in Lemma 2, Theorem 4, and Algorithm 2.
>
> > Intuitive explanation of privacy loss
>
> Under the definition of differential privacy, the privacy loss is measured based on how output distribution differs as input changes.
> Note that in our mechanism, the bin index $j$ is only used when computing the output, but not revealed in the output. Thus, it does not incur extra privacy loss.
>
> > The distribution of X in Theorem 2
>
> In Theorem 2, the upper bound holds for each input $X=x$, while $X$ can follow any distribution.
>
> > Larger $m$ for ERM mechanism
>
> Here are experimental results of ERM and RQM when $m$ = 8, input $x \in [-1, 1]$, privacy budget $\epsilon=1$. RQM can achieve a Mean Absolute Error (MAE) of 2.091, with parameters $q=0.047, \delta=1.5$, and uniformly distributed bins. ERM can achieve a lower error of 2.037, with parameters $\gamma=0.879, \delta=2.5$, and bins: [-3.5, -2.8, -2.1, -1.4,  1.4,  2.1,  2.8,  3.5].

---

### Official Review · Reviewer_aP86 · 2024-03-23

**Q2-1 Originality-Novelty:** 4
**Q2-2 Correctness-Technical Quality:** 3
**Q2-5 Clarity Of Writing:** 3

**Q1 Summary And Contributions:**

This paper aims to provide DP randomized quantization for releasing discrete-valued data. This paper proposes a novel randomized quantization mechanism with discrete, unbiased outputs under the DP guarantee. It provides a general formulation of the randomized quantization method, with fixed bins. The paper summarized existing quantization mechanisms as their special cases. The paper further provided an optimal unbiased discretization mechanism with minimum MAE. Simple numerical results show that the proposed mechanism achieves the minimal MAE among existing randomized quantization mechanisms.

**Q2-3 Extent To Which Claims Are Supported By Evidence:**

4: Excellent: all claims are supported by very convincing evidence (in the form of comprehensive experimental evaluation, rigorous mathematical proofs, detailed (pseudo-)code, precise references, well-motivated and realistic assumptions) and the authors deliver what they promise.

**Q2-4 Reproducibility:**

4: Excellent: key resources (e.g. proofs, code, data) are available and key details (e.g. proof sketches, experimental setup) are comprehensively described for competent researchers to confidently and easily reproduce the main results.

**Q3 Main Strengths:**

Novelty: this paper provides a unified formulation for randomized quantization mechanisms. It provides a novel formulation to optimize the mechanism to achieve minimal MAE under privacy and unbiasedness constraints.

Soundness: the derivation of the problem formulation is clear. It theoretically shows what constraints the quantization mechanism should satisfy for privacy constraints.

Clarity: the experiment results comparing the optimal mechanism with existing mechanisms are clear and easy to understand.

**Q4 Main Weakness:**

1. Restriction on the algorithm:
  1. It is only used for the scalar case. All the analyses in the paper are for scalar input (or entry-wise discretization). Extension to vector-based quantization should be non-trivial.
  2. It requires a known distribution of input. The optimization problem takes an expectation form over the distribution of $x$. However, for practical applications, the distribution of $x$ might be unknown (e.g., for DPSGD in the experiment). In such a case, there might not be a well-defined problem formulation.
  3. It has a fixed size of the bins. The optimization constraints and problem are based on pre-defined bin sizes and values. Dynamic bins and quantizations have also been used, e.g., [R1]. It is unclear how the algorithm generalizes to such settings.
  4. Biased quantization. There should be a tradeoff between bias, deviation, and privacy for the DP mechanism. The author only provides a tradeoff between privacy and deviation and keeps the bias as zero. Could the author extend the mechanism to possibly biased quantization?

2. Limited experiments: Most of the experiments assume that the parameters have uniform distribution over the range. It would be good if other distributions (e.g., truncated Gaussian) were considered.


[R1] Zhou, Y., Moosavi-Dezfooli, S. M., Cheung, N. M., & Frossard, P. (2018, April). Adaptive quantization for deep neural network. In Proceedings of the AAAI Conference on Artificial Intelligence (Vol. 32, No. 1).

**Q5 Detailed Comments To The Authors:**

It will be great to see how the algorithm generalizes to high-dimensional quantization or how to op[optimize the bins.

More sets of experiments under different distributions would strengthen the paper.

I noticed that the experiments use different bins for different quantization mechanisms. Please discuss why these bins are selected.

**Q9 Complying With Reviewing Instructions:**

Yes

---

> ### Author Rebuttal · Authors · 2024-04-09
>
> Thanks for your comments. Here are our responses point by point:
>
> > High-dimensional quantization
>
> Our method can be extended to higher-dimensional quantization with the same method as in MVU[R3]. Specifically, for any $d$-dimensional input vector $\textbf{x}=(\textbf{x}_1, \cdots, \textbf{x}_d)$ with $L_2$ norm bounded by diameter $\Delta$, we map the input vector $\textbf{x}$ to $\mathcal{M}_d(\textbf{x})=(\mathcal{M}^{\prime}(\textbf{x}_1), \cdots, \mathcal{M}^{\prime}(\textbf{x}_d))$. Here, the mechanism $\mathcal{M}^{\prime}$ quantize the scalar in each coordinate and needs to satisfy $\epsilon$-metric DP, a variant of $\epsilon$-DP that requires the following holds for any two inputs $x, x^{\prime}$ and any set of possible outputs $S \subseteq$ Range($\mathcal{M}$): $$\Pr(\mathcal{M}^{\prime}(x) \in S) \leq e^{\epsilon d(x, x^{\prime})} \Pr(\mathcal{M}^{\prime}(x^{\prime}) \in S),$$ where $d(x, x^{\prime})=|x - x^{\prime}|^2$. Since Lemma 6 in [R3] has shown that the mechanism $\mathcal{M}_d$ generated by $\epsilon$-metric DP $\mathcal{M}^{\prime}$ is $\epsilon \Delta^2$-DP and unbiased, we can directly use our method to find the optimal parameters of $\mathcal{M}^{\prime}$ (under new privacy constraints specified by $\epsilon$-metric DP).
>
> > Unknown distribution of input $x$
>
> Our mechanism can be generalized to non-uniform or partially known distribution of input $X$, and we can assign asymmetric bins to capture the pattern of the distribution. Specifically, when input distribution is unknown, we can first collect samples of inputs. For any given bin values, we can estimate the probability that inputs fall into the intervals between different pairs of neighboring bins. Then we can extend Lemma 1 and Theorem 3 to find a linear upper bound on the error as follows:
> $$\mathbb{E}(|\mathcal{M}(X)-X|)
>     \leq \sum_{i=s-1}^{t}  (\zeta_{m-i}^{(r)}+B_{i+1}-B_{i}+\zeta_{i}^{(l)}) \int_{\max(B_i, -c)}^{\min(B_{i+1}, c)} f_X(x) dx,$$
> where $\zeta_i^{(l)}$, $\zeta_{m-i}^{(r)}$ are similar to the definition of $\zeta_n$ in Section 4.2, except that they are defined over bins on the left and right side of $x$, respectively; $f_X(x)$ is the probability density function of $X$; $B_s<B_t$ are bins in $[-c,c]$ closest to $-c$ and $c$, respectively. Note that this upper bound only depends on density $f_X(x)$ through the integral $\Pr(B_i \leq X < B_{i+1}) = \int_{B_i}^{B_{i+1}} f_X(x)dx$, which is easier to know (compared to density itself) and can be estimated from samples. Therefore, we can minimize this upper bound with privacy constraints through linear programming.
>
> > Dynamic bins
>
> Our method can also be used when different inputs require quantization mechanisms with different hyperparameters. We illustrate this using [R1,R2] as examples. Specifically, [R1] decides the optimal bit-width of quantization for different layers of deep neural networks. [R2] uses dynamic quantization to decide the clipping range of activation values during runtime. In both cases, after hyperparameters (e.g., number of bins, clipping range) are decided, we can directly use our algorithm to find the optimal parameters with given hyperparameters.
>
> > Biased quantization
>
> Our mechanism can be extended to biased quantization. Instead of randomly outputting $B_l$ or $B_r$ and enforcing unbiasedness according to Eq. (1) in Section 3.2, we can use exponential mechanism with parameter $\epsilon$ to output $B_l$ or $B_r$, with score function being the negative distance between input and output bins. This mechanism induces biased output but reduces privacy loss.
>
> > Experiment under different distributions
>
> We test our mechanism under truncated Gaussian distribution. When $m=4$, input $X$ follows Gaussian distribution with $\mu = 0.5, \sigma = 0.1$ and is truncated by $[-1, 1]$, our OPTM with asymmetric bins and optimized with sample distribution has a Mean Absolute Error of 1.76, lower than 1.94 of MVU and 2.03 of RQM. This indicates that OPTM which is specifically optimized for input distribution can have advantages over other mechanisms which are agnostic to the input distribution.
>
> > Reason of using different bins
>
> The reason that we use different bins for different mechanisms is mainly due to the design of these mechanisms. RQM uses uniformly-distributed bins. MVU uses non-linear optimization to find the optimal bin values. For ERM and OPTM, which are proposed in this paper and enable non-uniformly distributed bins, we use grid search to find hyperparameters with optimal performance.
>
> [R1] Zhou, Yiren, et al. "Adaptive quantization for deep neural network." Proceedings of the AAAI Conference on Artificial Intelligence. Vol. 32. No. 1. 2018.
>
> [R2] So, Junhyuk, et al. "Temporal dynamic quantization for diffusion models." Advances in Neural Information Processing Systems 36 (2024).
>
> [R3] Chaudhuri, Kamalika, Chuan Guo, and Mike Rabbat. "Privacy-aware compression for federated data analysis." Uncertainty in Artificial Intelligence. PMLR, 2022.

---

### Official Review · Reviewer_Kgb2 · 2024-03-27

**Q2-1 Originality-Novelty:** 2
**Q2-2 Correctness-Technical Quality:** 2
**Q2-5 Clarity Of Writing:** 2

**Q1 Summary And Contributions:**

This paper presents a differential privacy mechanism for discrete data. This has been an ongoing challenge in the field which is normally focuses on continuous data. An interesting linear programming approach is presented, along with theoretical properties, and empirical results.

**Q2-3 Extent To Which Claims Are Supported By Evidence:**

2: Fair: the main claims are somewhat supported by evidence (but the experimental evaluation may be weak, or does not match entirely with the claims, important baselines may be missing, proofs contain important ideas but lack rigor, algorithmic details are only discussed superficially, references are imprecise, assumptions are not sufficiently motivated or explicated, etc.).

**Q2-4 Reproducibility:**

2: Fair: key resources (e.g. proofs, code, data) are unavailable but key details (e.g. proof sketches, experimental setup) are sufficiently well-described for an expert to confidently reproduce the main results.

**Q3 Main Strengths:**

The paper presents a novel approach to a challenging problem and presents rigorous analysis of it along with a good evaluation.

**Q4 Main Weakness:**

The paper is quite technical and is probably readable by those who have a good understanding of differential privacy. An early example of what is done in the paper would be very helpful so readers who are not expert could get something from the paper.

There is no conclusion section.

**Q5 Detailed Comments To The Authors:**

Differential privacy is not a topic I am particularly expert in, however I am aware of the challenges of discrete domains. This paper seems technical strong, and generally well-written. I would have liked to have seen an opening example to illustrate the approach, and maybe a detailed one once the technique was presented. Overall I think this is a very nice paper.

**Q9 Complying With Reviewing Instructions:**

Yes

---

> ### Author Rebuttal · Authors · 2024-04-09
>
> Thanks for your comments. We will add a conclusion section and the following opening example to the revised manuscript.
>
> > Opening example
>
> A quantization mechanism $\mathcal{M}$ maps the continuous input $X$ to discrete output bins, i.e., $\mathcal{M}(X) \in {B_i}, i \in [m]$, where $m$ is the total number of output bins. The goal of this paper is to design the parameters of the mechanism such that the
> outputs $\mathcal{M}(X)$ are unbiased and satisfy differential privacy. Consider an example where $m=4$ and $x \in [B_2, B_3)$. Our mechanism finds $\mathcal{M}(x)$ in the following way: 1) it randomly selects one bin on the left of $x$ (e.g., $B_1$) and another bin on the right (e.g., $B_3$) according to a selection distribution, which are the parameters we need to tune; 2) it then randomly outputs either $B_1$ or $B_3$ while preserving unbiasedness. Our paper will formulate a linear program to find the optimal parameters of this mechanism.

---

### Meta-Review · Area_Chair_aZR8 · 2024-04-14

This paper proposes a randomized quantization mechanism with discrete, unbiased outputs under the DP guarantee. It proposes a linear programming approach to find the optimal parameters of the mechanism for a fixed privacy budget.

The paper is clearly written and the problem is interesting and important. The proposed method generalizes previous mechanisms.  However, it has some limitations on the family of data distributions, the bin sizes and the complexity of the optimization step. The extension for unknown, non-uniform distributions and dynamic bin sizes should be explained in the paper. The important limitation due to the grid search step should be reported in a final section of the paper, which is missing.